# Characterization of CrufCSP1 and Its Potential Involvement in Host Location by *Cotesia ruficrus* (Hymenoptera: Braconidae), an Indigenous Parasitoid of *Spodoptera frugiperda* (Lepidoptera: Noctuidae) in China

**DOI:** 10.3390/insects14120920

**Published:** 2023-12-01

**Authors:** Kai-Ru Han, Wen-Wen Wang, Wen-Qin Yang, Xian Li, Tong-Xian Liu, Shi-Ze Zhang

**Affiliations:** 1State Key Laboratory of Crop Stress Biology for Arid Areas, College of Plant Protection, Northwest A&F University, Yangling 712100, China; hankairu@nwafu.edu.cn (K.-R.H.); wangwenwen@nwafu.edu.cn (W.-W.W.); yangwq928@163.com (W.-Q.Y.); lixian0728@126.com (X.L.); 2Institute of Entomology, College of Agriculture, Guizhou University, Guiyang 550025, China; tx.liu@gzu.edu.cn

**Keywords:** chemosensory protein, *Cotesia ruficrus*, expression profiles, fluorescence binding assay, molecular docking, *Spodoptera frugiperda*, biological control

## Abstract

**Simple Summary:**

A previous study found that an indigenous endoparasitic wasp *Cotesia ruficrus*, has a good control effect on the invasive pest *Spodoptera frugiperda*. Understanding the molecular mechanism of host recognition of *C. ruficrus* would help improve its biological control effect. Chemoreceptor proteins (CSPs) play a crucial role in insect olfactory perception. The CrufCSP1 protein was obtained via prokaryotic expression, and its binding properties were evaluated using fluorescence binding assays. Molecular docking was used to verify the function of CrufCSP1. The findings revealed that CrufCSP1 has the ability to bind with seven host-related compounds. This information serves as a crucial reference for developing natural enemy attractants for pest control.

**Abstract:**

Chemosensory proteins (CSPs) are a class of soluble proteins that facilitate the recognition of chemical signals in insects. While CSP genes have been identified in many insect species, studies investigating their function remain limited. *Cotesia ruficrus* (Hymenoptera: Braconidae) holds promise as an indigenous biological control agent for managing the invasive pest *Spodoptera frugiperda* (Lepidoptera: Noctuidae) in China. This study aimed to shed light on the gene expression, ligand binding, and molecular docking of CrufCSP1 in *C. ruficrus*. A RT-qPCR analysis revealed that the expression of CrufCSP1 was higher in the wings, with male adults exhibiting significantly higher relative expression levels than other developmental stages. A fluorescence competitive binding analysis further demonstrated that CrufCSP1 has a high binding ability with several host-related volatiles, with *trans*-2-hexenal, octanal, and benzaldehyde showing the strongest affinity to CrufCSP1. A molecular docking analysis indicated that specific amino acid residues (Phe24, Asp25, Thr53, and Lys81) of CrufCSP1 can bind to these specific ligands. Together, these findings suggest that CrufCSP1 may play a crucial role in the process of *C. ruficrus* locating hosts. This knowledge can contribute to the development of more efficient and eco-friendly strategies for protecting crops and managing pests.

## 1. Introduction

In complex natural environments, natural enemies primarily rely on their olfactory systems to search for and recognize hosts by detecting host-related chemical substances, and these substances include herbivore-induced volatiles and pheromones [1,2,3]. The process of insect olfaction, from sensation to transmission, is mediated by various olfaction-related proteins, such as odorant-binding proteins (OBPs), chemosensory proteins (CSPs), odorant receptors (ORs), and ionotropic receptors (IRs) [4,5].

CSPs, initially referred to as OS-D or A10, are a critical class of small water-soluble proteins [6]. When insects are exposed to odorants in the external environment, some odor molecules can enter their bodies through the olfactory pores of the cuticle [7]. At this point, CSPs bind to hydrophobic odorants and serve as carriers, transporting them to olfactory sensory neurons (OSNs) [8]. The transported odorant signals are then transmitted to ORs embedded in the neuronal membrane, activating them [9]. This chemical-to-electrical signal conversion stimulates behavioral responses in insects [10]. CSPs can interact with different types of hydrophobic odorants due to their specific structures and affinities, and this enables insects to perceive and discriminate various chemical substances [11]. Additionally, CSPs are widely distributed in insect tissues and exhibit a broad expression pattern throughout their bodies. For instance, certain Hymenoptera species specifically express CSPs in their antennae [12]. In *Spodoptera exigua*, CSPs are expressed to varying degrees in the antennae, heads, thoraxes, abdomens, legs, and wings [13]. Similarly, *Adelphocoris suturalis* expresses CSPs in the antennae and bodies of both male and female adults [14]. The diverse expression patterns of CSPs are associated with various functions reported in different insect species. These functions include host localization in *Microplitis mediator* [15], sex pheromone detection in *Helicoverpa armigera* [16], metamorphosis and molting in *Drosophila* [17], leg regeneration in *Periplaneta americana* [18], and insecticide resistance in *Plutella xylostella* [19]. By studying the function of CSPs, we can enhance our understanding of their potential roles and promote the development of comprehensive control methods that target CSPs as molecular targets.

The fall armyworm (FAW), *Spodoptera frugiperda* (Lepidoptera: Noctuidae), is a polyphagous pest that feeds on over 350 plant species worldwide [20]. In China, FAW was first reported in January 2019 in a corn field in Yunnan Province, has now spread to 27 provinces, and is considered a serious threat to Chinese agricultural production and food security [21,22]. Currently, chemical control is the primary method used to manage *S. frugiperda* populations. However, the development of resistance and the high cost associated with chemical control have reduced its effectiveness and increased its costs. Therefore, it is important to develop alternative sustainable control technologies such as biological control for managing this insect. Biological control, particularly using parasitoids as primary agents, combined with other control measures, has been shown to reduce costs, minimize agricultural non-point source pollution, and achieve long-term and effective control [23]. Several studies demonstrated that parasitic wasps can limit the population growth of *S. frugiperda* to some extent [24,25]. The parasitoid species *Cotesia ruficrus* (Hymenoptera: Braconidae) is an endoparasitic wasp that can parasitize the larvae of several pests in Noctuidae and Pyralidae families [26,27]. In July 2020, our research team conducted a field investigation in Yangling, Shaanxi Province, China, and found *C. ruficrus* to be the dominant species parasitizing *S. frugiperda* in corn fields, and this is the first report of the indigenous parasitoid *C. ruficrus* parasitizing FAW in China [28]. Our previous work showed that *C. ruficrus* primarily parasitizes first-third instar larvae of *S. frugiperda* with a parasitism rate of 34–45% and greatly reduces their appetite, which implies that *C. ruficrus* is a promising local candidate for integrated pest management programs targeting the invasive pest *S. frugiperda* [29].

This study is based on the antennal transcriptome data of *C. ruficrus*, and three CSP genes (CrufCSP1, CrufCSP2, and CrufCSP3) of *C. ruficrus* were identified. Moreover, our preliminary experiments showed that CrufCSP1 was highly expressed in male, while CrufCSP2 and CrufCSP3 were highly expressed in female. Therefore, this paper only reports the results of CrufCSP1 (Accession No.: OR485561), and the functions of the other two CSP genes will be reported in another paper. The RT-PCR method was used to clone the CrufCSP1 gene, and its expression profiles in different developmental stages and tissues were determined. To further investigate the olfactory function of CrufCSP1, we conducted fluorescence binding assays to characterize its binding affinity with host-related volatiles. Additionally, we predicted the protein-binding sites via molecular docking. These comprehensive analyses provide valuable insights into the olfactory function of *C. ruficrus* and offer potential environmentally friendly techniques for controlling *S. frugiperda*. By understanding the molecular mechanisms underlying the olfactory function of *C. ruficrus*, we can develop more targeted and effective control strategies for managing *S. frugiperda*. These findings pave the way for the development of sustainable pest management strategies.

## 2. Materials and Methods

### 2.1. Insect Rearing

The collection and rearing methods for the *S. frugiperda* and *C. ruficrus* colonies were based on the study conducted by He et al. [29]. In brief, *S. frugiperda* larvae were reared on corn (*Zea mays* L. var. Shandan 636), and *C. ruficrus* parasitized second-third instar larvae of *S. frugiperda* in a controlled artificial climate greenhouse with the temperature maintained at 26 ± 1 °C and the relative humidity at 70 ± 5%, with a photoperiod of 14L:10D. The parasitized larvae of *S. frugiperda* were collected individually and placed into 3 × 3 cm petri dishes containing fresh corn leaves. During this process, parasitic wasp cocoons were carefully collected and transferred to petri dishes. Once the parasitoids emerged as adults, they were provided with 10% honey water for nutrition.

### 2.2. RNA Extraction, cDNA Synthesis, and Gene Cloning

We collected various samples of *C. ruficrus* to extract total RNA for further analysis. A total of 25 third larvae, 20 pupae (3 days old), 30 unmated male or female adults (1 day old), and 20 mated male or female adults (1, 3, 5, and 7 days old) were used for RNA extraction. To obtain different tissues of *C. ruficrus*, we collected antennae (n = 150), heads without antennae (n = 150), thoraxes (n = 100), abdomens (n = 100), legs (n = 150), and wings (n = 150) from unmated 1-day-old male and female adults using a scalpel and forceps. Each treatment was represented by three biological replicates. Total RNA was extracted from these samples using TRIzol reagent (TaKaRa, Dalian, China) according to the manufacturer’s instructions. The quality and quantity of the extracted RNA were assessed using a NanoDrop 2000 spectrophotometer (Thermo Scientific, Wilmington, DE, USA). The cDNA synthesis was performed using the Prime Script^TM^ RT Reagent Kit with gDNA Eraser (TaKaRa, Dalian, China), and the synthesized cDNA samples were stored at −20°C for subsequent experiments and analysis.

The primers used to validate the CrufCSP1 sequence are listed in Table 1. The RT-PCR reactions were performed according to the following conditions: polymerase activation at 95 °C for 5 min, followed by 34 cycles of denaturation at 95 °C for 30 s, annealing at 59 °C for 30 s, and extension at 72 °C for 30 s. A final extension step was conducted at 72 °C for 10 min to ensure the completeness of the PCR reaction. After amplification, the PCR products were purified and ligated into the PMD19-T vector. The recombinant vectors were then transformed into DH5α competent cells (TaKaRa, Dalian, China) and sequenced.

### 2.3. Identification, Sequence Analysis, and Phylogenetic Tree Construction of CrufCSP1

The open reading frame (ORF) of CrufCSP1 was predicted using the online software ORFfinder (https://www.ncbi.nlm.nih.gov/orffinder/ (accessed on 12 March 2023)). The EditSeq in Lasergene software was used to predict the amino acid sequence encoded by CrufCSP1. The SignalIP program server (https://services.healthtech.dtu.dk/service.php?SignalP-6.0 (accessed on 12 March 2023)) was employed to identify potential N-terminal signal peptides of CrufCSP1. ExPASy (https://www.expasy.org/ (accessed on 15 March 2023)) was utilized to predict the molecular weight and isoelectric point of CrufCSP1. To compare CrufCSP1 with CSPs from other Hymenoptera insects, sequence alignment was performed using DNAMAN 9.0 software (Lynnon Biosoft, San Ramon, CA, USA). Based on the nucleotide sequences of CrufCSP1 and other insect CSPs, MEGA 11 software was employed to construct a phylogenetic tree. The neighbor-joining method was used for tree construction, and the bootstrap procedure with 1000 replicates was applied for statistical support. To enhance the clarity and visualization of the phylogenetic tree, the resulting tree was color-coded according to taxonomy using the tvBOT tool (https://www.chiplot.online/ (accessed on 10 April 2023)).

### 2.4. Expression of CrufCSP1 Gene

The primers that were designed based on the cDNA sequences were used for RT-qPCR (Table 1). The β-actin gene was chosen as a reference gene for the expression levels of CrufCSP1 in different life stages and tissues. The RT-qPCR reactions were conducted using TB Green^®^ Premix Ex TaqTM II (TaKaRa, Dalian, China) and the LightCycler480 real-time PCR system (Roche Diagnostics, Switzerland). The reaction conditions consisted of an initial denaturation step at 95 °C for 30 s, followed by 40 cycles of denaturation at 95 °C for 10 s, annealing at 55 °C for 30 s, and extension at 72 °C for 40 s. For each sample, three biological replicates and technical replicates were performed. The 2^−ΔΔCT^ method was used to determine the mRNA expression levels of CrufCSP1.

The data were first checked for normality of distribution and homogeneity of variance. The relative expression levels of CrufCSP1 between different developmental stages and tissues of both female and male specimens were analyzed by one-way analysis of variance (ANOVA), followed by Tukey’s multiple comparison test when means had significant differences at *p* < 0.05. The expressions of CrufCSP1 between male and female were analyzed by Student’s *t*-test. Data were analyzed by using statistical software package SPSS 22.0 and shown as mean ± standard error.

### 2.5. Cloning and Construction of Recombinant Plasmids

Table 1 lists the primers designed using Primer Premier 5.0, which include BamHI and Xhol restriction enzyme sites. These primers were specifically designed to enable the amplification of the coding region of CrufCSP1. After PCR amplification, the resulting products were purified and ligated into the pMD19-T vector. The ligation product was then introduced into DH5α cells (TaKaRa, Dalian, China) via a transformation process. Subsequently, the positive clones were selected. The chosen clones were cultured in Luria–Bertani (LB) medium and then sequenced. The pMD19-T plasmid carrying the CrufCSP1 insert was digested using BamHI and Xhol restriction enzymes (TaKaRa, Dalian, China) at 37 °C for 3 h and then ligated into the expression vector pET30a (+). The recombinant plasmid was subsequently transformed into *Escherichia coli* DH5α cells. To proceed with protein expression, the correctly transformed plasmid was further introduced into *E. coli* BL21 (DE3) cells. To ensure the stability and correctness of the recombinant plasmid, at least 6 single clones were cultured in LB medium. Positive colonies were identified and confirmed via sequencing, ensuring the reliability and accuracy of the subsequent experiments.

### 2.6. Prokaryotic Expression and Purification of CrufCSP1

The positive colonies were diluted in liquid LB medium to a volume of 800 mL and then incubated at 37 °C. The cells were allowed to grow until they reached an optical density (OD6_00_) value between 0.4 and 0.8. To induce protein expression, isopropyl β-D-1-thiogalactopyranoside (IPTG) was added to the culture at a final concentration of 1 mM. The culture was then further incubated at 37 °C for at least 6 h. After the incubation period, the cells were collected via centrifugation at 12,000× *g* for 20 min at 4 °C. The resulting cell pellet was subsequently sonicated in ice, followed by another round of centrifugation. Upon analysis, the CrufCSP1 protein was expressed in inclusion bodies. To purify the target protein, a series of denaturation and refolding steps was employed. The inclusion bodies were dissolved using a solution containing 50 mM of tris buffer (pH 6.8) and 0.2% Triton X-100. Subsequently, an 8 M solution of urea was added to facilitate dissolution. For refolding, a solution consisting of 0.5 M of NaOH with a cystine concentration of 5 mM was utilized. Additionally, a solution containing 100 mM of Tris-HCl (pH 8.0) with a cysteine concentration of 5 mM was employed. The refolded protein was collected and purified with BeyoGold™ His-tag Purification Resin (Beyotime, Shanghai, China). The purity of the purified CrufCSP1 protein was assessed with a 15% SDS-PAGE analysis. Additionally, the protein concentration was determined using the BCA protein quantification kit (Beyotime, Shanghai, China).

### 2.7. Fluorescence Binding Assays

The binding affinity between CrufCSP1 and the selected volatile compounds was determined using an F-7000 fluorescence spectrophotometer (Hitachi, Tokyo, Japan). N-phenyl-1-naphthylamine (1-NPN), a fluorescence probe, was used to assess the binding affinity. A total of 24 volatiles, i.e., alcohols (heptanol, undecanol, tetradecanol, *cis*-3-hexen-1-ol, 1,14-tetradecanediol, 2-hexanol, phytol, and linalool), aldehydes (hexanal, heptanal, octanal, nonanal, decanal, dodecanal, pentadecanal, benzaldehyde, *p*-methyl benzaldehyde, and *trans*-2-hexenal), and others (2-hydroxy-5-methylacetophenone, 2-heptanone, palmitic acid, *N*,*N*-dimethylcaprylamide, DMNT, and *cis*-3-hexenyl acetate), were selected for the fluorescence binding assay with CrufCSP1. Of these compounds, our research group previously identified undecanol, tetradecanol, 2-hexanol, 1,14-tetradecanediol, phytol, dodecanal, pentadecanal, *p*-methyl benzaldehyde, palmitic acid, 2-hydroxy-5-methylacetophenone, and *N*,*N*-dimethylcaprylamide in the larvae of *S. frugiperda*. The remaining compounds were chosen based on previous studies that identified them as corn volatiles [30,31].

To prepare the tested chemicals, all were dissolved in chromatographic grade methanol to achieve a 100 mM solution, which was then stored at −20 °C. Before use, this solution was diluted to 1 mM. CrufCSP1 was dissolved in 20 mM of tris-HCl (pH 7.4) and then diluted to a 2 μM working solution. To ensure the binding affinity of 1-NPN to the protein, the CrufCSP1 solution was titrated with 1-NPN with a concentration range of 2 to 20 μM. For the fluorescence competitive binding assays, each ligand was added to a mixture containing CrufCSP1 (2 μM) and 1-NPN (2 μM) at concentrations ranging from 0 to 30 μM. The maximum fluorescence intensity was then recorded and plotted against the ligand concentration based on the average of the three repeats. The dissociation constant (Kd) for binding between CrufCSP1 and 1-NPN was calculated using the Scatchard linear regressive equation in the software GraphPad Prism 8.0 (GraphPad Software Inc., La Jolla, CA, USA). Additionally, the ligand-binding affinity (Ki) was calculated utilizing the following equation: $$K_{i}=[{IC}_{50}]/(1+[1-NPN]/K_{1-NPN})$$
where IC_50_ represents the maximum concentration at which the ligand replaces 50% of the 1 − NPN fluorescence, [1 − NPN] denotes the free concentration of 1-NPN, and K_1-NPN_ represents the dissociation constant of the protein/1 − NPN complex [32].

### 2.8. Three-Dimensional Structure and Molecular Docking of CrufCSP1

The 3D modeling of CrufCSP1 was performed using the online program SWISS-MODEL (https://swissmodel.expasy.org/ (accessed on 5 June 2023)), with CSPMbraA6 (PDB ID: 1N8U) as a template, as described by Tomaselli et al. [33]. To evaluate the quality of the final 3D model, Structure Analysis and Verification Servers (http://services.mbi.ucla.edu/SAVES/ (accessed on 5 June 2023)), including Procheck and ERRAT, were utilized. Based on the results of the fluorescence binding assays, the ligands including *trans*-2-hexenal, octanal, benzaldehyde, *N*,*N*-dimethylcaprylamide, hexanal, linalool, and 2-hydroxy-5-methylacetophenone (K_i_ < 20 μM) were selected for further analysis. The 3D structures of these ligands were downloaded in SDF format from PubChem (https://pubchem.ncbi.nlm.nih.gov/ (accessed on 7 June 2023)) and then converted to PDBQT format using Open Babel.

The protein-ligand complexes between CrufCSP1 and the selected molecules were constructed using Autodock (version 1.5.6). The CrufCSP1 receptor and ligand molecules were processed and saved in PDBQT format after undergoing necessary steps such as dehydration, hydrogenation, and charge calculation, which were performed using Mgtools 1.5.6. Subsequently, Autodock Vina 1.1.2 was utilized to conduct the docking simulations of the ligands with the receptor. The conformation with the lowest binding energy was selected as the optimal binding mode of each complex. For the visual analysis of the protein 2D and 3D structures, Discovery Studio 2019 Client and PyMOL version 2.3.0 were used.

## 3. Results

### 3.1. Identification and Phylogenetic Analysis of CrufCSP1 in C. ruficrus

The analysis of the CrufCSP1 sequence showed that its open reading frame (ORF) consisted of 249 bp, encoding 82 amino acids. The molecular weight of CrufCSP1 was calculated to be 9.04 kDa, and its isoelectric point was 8.32. Notably, no signal peptide was predicted at the N-terminus of CrufCSP1 (Figure 1A). Furthermore, the sequence alignment between CrufCSP1 and nine other CSPs from Hymenoptera insects demonstrated the presence of four conserved cysteine residues within CrufCSP1, exhibiting the pattern C1-X6-8-C2-X16-21-C3-X2-C4, where X represents any amino acid except cysteine (Figure 1B). Moreover, to investigate the functional relationship of CrufCSP1, a phylogenetic tree was constructed using the nucleotide sequences of 58 CSP genes obtained from the NCBI database. Based on the morphological characteristics of the resulting phylogenetic tree, CrufCSP1 clustered together with the CSP22 from *Cnaphalocrocis medinalis*, which suggests that CrufCSP1 may play a role in host recognition (Figure 2).

### 3.2. Expression Profile Analysis of CrufCSP1

The qPCR results show that the expression of CrufCSP1 was significantly higher in male adults compared with female adults, larvae, and pupae (F = 1139.277, *p* < 0.001) (Figure 3A). Moreover, the temporal analysis of the two sexes revealed that the expression level of CrufCSP1 in males and females was highest on the first day post-eclosion and then gradually decreased with time after emergence (Figure 3B; ♂: F = 61.66, *p* < 0.001; ♀: F = 8.907, *p* < 0.01). Moreover, the expression level of CrufCSP1 in males was significantly higher than that in females on the first day (t = 4.45, *p* < 0.05) and the third day (t = 4.35, *p* < 0.05). Notably, in both sexes, wings exhibited the highest expression level, while the distribution of CrufCSP1 showed varying degrees in antennae, heads, thoraxes, and legs (Figure 3C; ♂: F = 146.4, *p* < 0.001; ♀: F = 54.43, *p* < 0.001). Furthermore, the expression levels of CrufCSP1 in the heads (t = 4.13, *p* < 0.05), legs (t = 11.12, *p* < 0.01), and wings (t = 7.32, *p* < 0.01) of females were significantly higher than those in males, but the expression level in the abdomen was significantly higher in males than that in females (t = 16.71, *p* < 0.01).

### 3.3. Recombinant Protein Expression and Purification of CrufCSP1

The recombinant CrufCSP1 protein was efficiently produced and purified in *E. coli*, resulting in a concentration of 0.53 mg/mL. During the expression process, CrufCSP1 predominantly accumulated in insoluble bodies. The presence of the His-tag on the protein was confirmed via SDS-PAGE, which revealed a molecular weight of 15 kDa, which was consistent with the predicted results (Appendix A).

### 3.4. Fluorescence Binding Assays with CrufCSP1

To assess the ligand-binding capabilities of the purified CSP proteins, we initially measured the binding affinity between fluorescent probe 1-NPN and CrufCSP1 (Figure 4). CrufCSP1 exhibited a dissociation constant of 7.668 µM toward 1-NPN. Subsequently, to further investigate the function of the recombinant protein, we conducted a competitive fluorescence binding assay using 24 different compounds (Table 2). Among these, CrufCSP1 demonstrated a significantly higher affinity for seven tested compounds, detailed as follows: *trans*-2-hexenal (14.49 μM), octanal (15.04 μM), benzaldehyde (15.24 μM), *N*,*N*-dimethylcaprylamide (19.29 μM), hexanal (19.36 μM), linalool (19.80 μM), and 2-hydroxy-5-methylacetophenone (19.97 μM) (Figure 5). Conversely, CrufCSP1 displayed a weaker binding affinity (K_i_ > 20 μM) toward the remaining ligands. The binding curves of selected ligands are listed in Figure 5.

### 3.5. Three-Dimensional Model Structuring and Molecular Docking of CrufCSP1

To better understand the characteristics of CrufCSP1, we analyzed its sequence and generated a 3D structure (Figure 6). Our analysis revealed that CrufCSP1 shares 59% sequence similarity with CSPMbraA6 (Figure 6A). To assess the quality of the predicted model, we conducted an ERRAT analysis, which demonstrated that the error value of the model structure was 100% (Appendix A), suggesting that the non-bonding interactions between different atoms in the protein were reasonable. Furthermore, the Ramachandran plot indicated that 94.5% of the amino acids were in the most favorable regions (Appendix A), indicating the reliability of the predicted model. The 3D visual analysis revealed that CrufCSP1 is composed of four α-helices (α1–α4) which are located between residues Val29-Asn35 (α1), Lys36-Leu46 (α2), Ser54-Ala68 (α3), and Pro76-Glu80 (α4) (Figure 6A). These structural features stabilize the interior cavity of the protein, which has a volume of 1163 Å^3^ (Figure 6B).

To confirm the binding affinity of ligands with CrufCSP1, we carried out molecular docking and calculated the binding energy between CrufCSP1 and each ligand. The results demonstrate that the ligands bound to the pocket of CrufCSP1 with negative energy values, and the distance between potential interacting residues was less than 4 Å (Table 3). Figure 7 presents the optimal orientations of the bound ligands, the conformations of the seven ligands in the hydrophobic region, and the 2D interactions. Out of the seven compounds, six ligands could form hydrogen bonds with CrufCSP1 except for *N*,*N*-dimethylcaprylamide, which interacted with the protein via eight amino acid residues (Tyr20, Thr21, Gln58, Leu59, Lys62, Leu63, Ala66, and Gln78) (Figure 7D). The binding of CrufCSP1 to aldehyde compounds primarily relied on hydrogen bonds formed by Thr53 and Phe24, hydrophobic interactions (Leu38, and Tyr42), and van der Waals interactions (Ser22, Gly41, Gly56, and Glu58) (Figure 7A–C,E). We also observed that CrufCSP1 formed two hydrogen bonds with linalool located at the Asp25 and Lys81 amino acids (Figure 7F) and three hydrogen bonds with 2-hydroxy-5-methylacetophenone located at Phe24, Asp25, and Lys81 (Figure 7G).

## 4. Discussion

Parasitic wasps have been widely recognized as crucial contributors in biological control programs aimed at mitigating agricultural and forestry pests [34]. In recent years, *C. ruficrus* has emerged as a promising local biological control agent for managing the invasive pest *S. frugiperda* [28,29]. Consequently, gaining a comprehensive understanding of the molecular mechanisms that underlie the host location ability of *C. ruficrus* is of paramount importance for successfully combating this pest. In this study, we delved in depth into the gene expression, ligand binding, and molecular docking of CrufCSP1 in *C. ruficrus*. The research findings enhance our understanding of *C. ruficrus*’s molecular mechanisms in host recognition, thus offering novel avenues for the prevention and control of this destructive pest.

The chemosensory proteins of several parasitic wasp species such as *Scleroderma guani*, *M. mediator*, and *Cotesia vestalis* have been identified and studied [35,36]. In this study, our results show that CrufCSP1 of *C. ruficrus* and CSP22 of *C. medinalis* clustered together, and the expression of CrufCSP1 gene was highest in adults, significantly higher than in other developmental stages. Similar expression patterns have been observed in other insect species such as *Frankliniella occidentalis* CSP1 [11], *Encarsia formosa* CSP3 [37], and *Dendroctonus armandi* CSP1 [27]. Notably, the expression of CrufCSP1 in male *C. ruficrus* specimens was significantly higher than that in its female adults, which may be due to the need for the olfactory function to locate mates and perform other physiological activities. It is known that unmated female parasitoids lay unfertilized eggs that develop into males, and mated females can lay unfertilized eggs that will develop into males or fertilized ones that develop into females. Therefore, the key to successful mating and parasitism is how parasitoids utilize semiochemicals to quickly target and locate suitable mates and hosts [38]. Generally speaking, female parasitic wasps use herbivore-induced plant volatiles (HIPVs) and specific odors or non-volatile compounds released by hosts to find hosts for egg laying, while male parasitoids use sex pheromones to find females for courtship and mating [39,40]. Some gregarious and solitary braconid wasps may rely on pheromones in combination with HIPVs to locate suitable mates [41,42]. *Campoletis chlorideae* males use female-derived volatile sex pheromone to find females for courtship and mating [43]. The gregarious parasitoid *Cotesia glomerata* and the solitary parasitoids *Cotesia marginiventris* may rely on pheromones in combination with plant odors to locate suitable mates [44]. Our results suggest that CrufCSP1 in male *C. ruficrus* may have a similar role. Furthermore, CSPs were not only distributed in important olfactory organs, such as antennae, but also in the chemosensory organs of other body parts, such as wings, legs, and gonads. This indicates that CSPs with different distributions in the body may serve different functions. For example, BodoCSP1 in *Bradysia odoriphaga* is highly expressed in the legs of both female and male adults, participating in the perception of host plant volatile compounds [45]. Similarly, CSP4 and CSP1 are highly expressed in the external genitalia of male *Eogystia hippophaecolus*, suggesting their involvement in spousal positioning or mating activities [46]. In the present study, we found the highest expression of the CrufCSP1 gene in adult wings, consistent with the expression profile of the AcerCSP3 gene in bees [47]. However, further research is needed to explore whether CrufCSP1 expressed in wings has additional functions beyond olfaction.

We did not remove the His-tag label in the N-terminus of the CrufCSP1 protein during the fluorescence binding assays because previous studies showed that the presence of this label does not affect the protein’s binding activity [48]. Moreover, we conducted rigorous testing on the size and purity of CrufCSP1 before performing the fluorescence competitive binding assays to analyze its binding ability. Our findings reveal that 14 of the 24 tested volatile compounds showed a binding affinity for CrufCSP1, suggesting that an insect’s recognition of multiple volatile compounds is likely regulated by a single gene. Although CrufCSP1 demonstrated better binding capabilities with most of the corn volatiles than with body volatiles, additional olfactory proteins in parasitoids must exist that can recognize larval volatiles. Interestingly, all volatile organic compounds (VOCs) demonstrating affinity with CrufCSP1 were characterized by their polarity and a carbon structure of less than 10. However, some important compounds for natural enemies, such as DMNT and TMTT in their foraging behaviors for hosts and prey [49,50], were not bound by CrufCSP1. This observation suggests the likelihood of other CrufCSPs playing a role in the perception of crucial plant volatiles and apolar compounds. Subsequent studies should therefore aim to conduct more comprehensive research on OBPs and CSPs, integrating the binding spectra of multiple proteins, which could enable successful attraction of natural enemies l and increase of the parasitic rate in the field. Among the binding spectra of CrufCSP1, *trans*-2-hexenal, hexanal, and benzaldehyde are classic plant volatile components released by corn, while linalool is a component of HIPVs, and trace amounts of it can help plants attract natural enemies [51]. As an information transfer protein in the chemical-sensing system of *C. ruficrus*, CrufCSP1 plays a significant role as a scent molecule carrier when the parasitoid searches for nectar plants or HIPVs. Similarly, CSPs in other insects also contribute to the recognition of plant volatiles. BminCSP3, which is highly expressed in the antennae of *Bactrocera minax*, can bind citral [52], and MsepCSP14 can specifically regulate the recognition of R-(+)-limonene, linalool, farnesene, and nerolidol, volatile components derived from plants [53]. While we found that the compounds screened using fluorescence binding assays could bind to CrufCSP1, it remains necessary to conduct behavioral experiments to verify whether these compounds elicit a behavioral response and whether *C. ruficrus* exhibits a preference for attraction or repulsion between them.

To validate the ligand binding results, we performed 3D modeling and molecular docking. The 3D model of CrufCSP1 revealed that it consists of four α-helices and four conserved cysteine residues, forming a hydrophobic pocket. This is different from other insect CSPs, which typically have five [54,55] or six α-helices [56,57]. We speculate that the CrufCSP1 has fewer α-helices, as this might lead to the formation of a larger hydrophobic pocket in its spatial conformation, making it more conducive to the entry of odor molecules and the formation of stable complexes. Further research is needed to elucidate its role. Molecular docking analysis showed that *trans*-2-hexenal, the optimal ligand, binds to specific amino acid residues in the hydrophobic pocket of CrufCSP1, including Leu38, Gly41, Tyr42, Cys45, Cys52, Thr53, and Gly56. This suggests that CrufCSP1 recognizes *trans*-2-hexenal via these residues. Hydrogen bonding between odorant molecules and CSPs is crucial for specificity. Other CSPs have been found to form hydrogen bonds with specific amino acid residues, such as Asn66 in AcerCSP1 and Ser63 in AcerCSP2 in *Apis cerana* [47], Thr27 and Leu30 in GmolCSP8 in *Grapholita molesta* [58], and Glu122 and Lys121 in MsepCSP8 [59]. In the present study, the docking energy between CrufCSP1 and 2-hydroxy-5-methylacetophenone was the lowest, indicating a strong binding capability. Amino acid residues Phe24, Asp25, and Lys81 were found to closely interact with 2-hydroxy-5-methylacetophenone via hydrogen bonding, suggesting their important roles in the binding process. These findings support the idea that CrufCSP1 binds to ligands via specific recognition sites. The molecular docking results demonstrate that CrufCSP1 can recognize potential attractants such as *trans*-2-hexenal, octanal, benzaldehyde, *N*,*N*-dimethylcaprylamide, and hexanal. However, to understand the physiological functions of CrufCSP1, further studies using RNAi and behavioral assays are necessary.

## 5. Conclusions

CrufCSP1 showed a strong affinity for seven host-related volatiles, indicating its ability to identify and locate hosts. The results from the 3D modeling and molecular docking highlight the importance of specific amino acid residues, namely Thr53, Phe24, Asp25, and Lys81, in the binding process between CrufCSP1 and the volatiles. These findings suggest that CrufCSP1 plays a significant role in host recognition via its olfactory function. Understanding the olfaction targets of natural enemies such as CrufCSP1 is crucial for developing effective pest management strategies. Identifying and studying the specific volatiles that CrufCSP1 interacts with makes it possible to design targeted approaches for pest control or enhancing biological control methods. This knowledge can contribute to the development of more efficient and eco-friendly strategies for protecting crops and managing pests.

## Figures and Tables

**Figure 1 insects-14-00920-f001:**
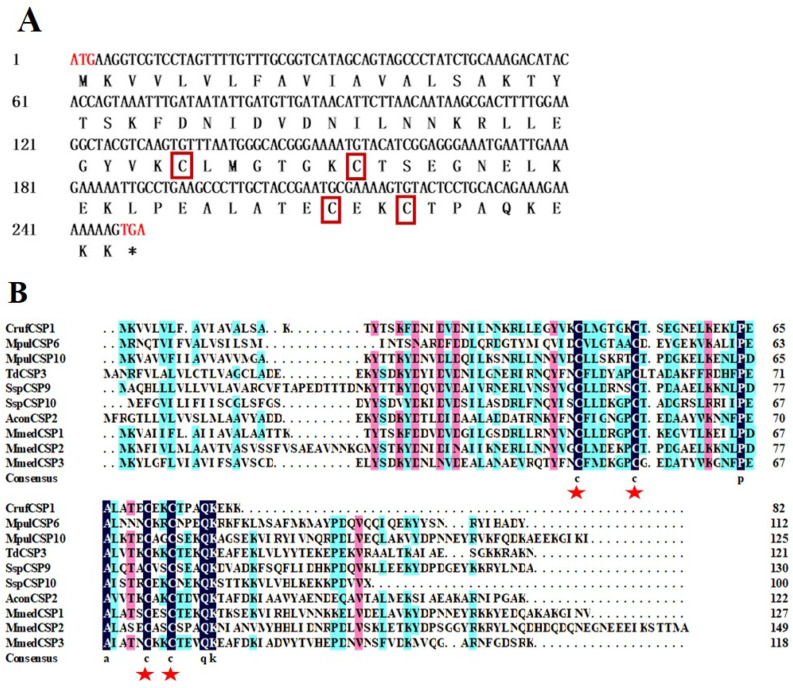
Nucleotide and deduced amino acid sequence of CrufCSP1 of *C. ruficrus* (**A**), and alignment of amino acid sequence of CrufCSP1 of *C. ruficrus* with other Hymenoptera insects (**B**). Note: Four conserved cysteines of CrufCSP1 are highlighted in red boxes, and the stop codon is indicated by a * in (**A**). Four conserved cysteines are also marked in red, and the degree of sequence similarity is represented as follows: black indicates 100%, pink indicates 80%, and sky blue indicates 60% in (**B**). The names and GenBank accession numbers of the sequences used for comparison are provided as follows: Cruf: *Cotesia ruficrus*, Mpul: *Meteorus pulchricornis*, Td: *Trichogramma dendrolimi*, Ssp: *Sclerodermus* sp., Acon: *Aulacocentrum confusum*, and Mmed: *M. mediator*.

**Figure 2 insects-14-00920-f002:**
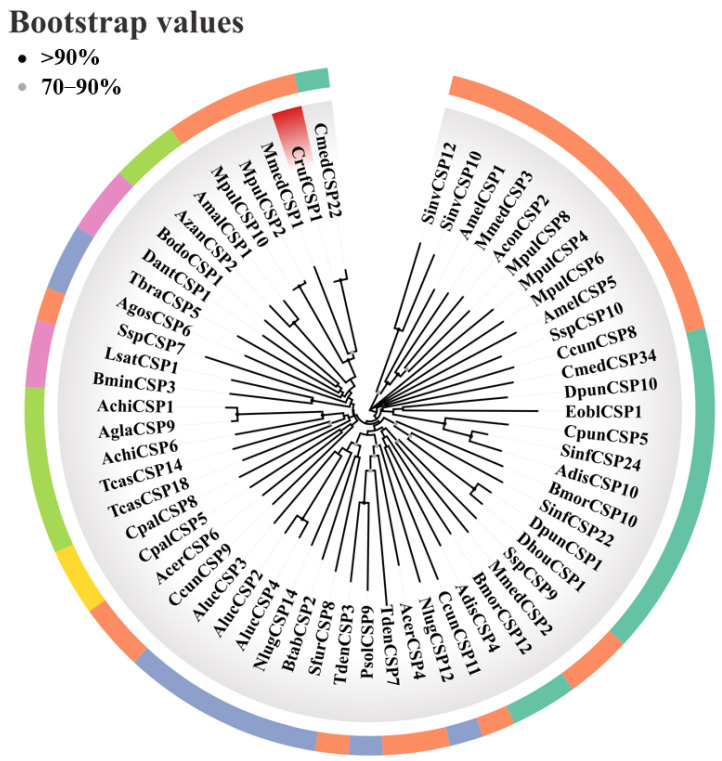
Phylogenetic tree of CrufCSP1 with 58 chemosensory proteins from other insect species. CrufCSP1 is highlighted with a red box, while other insect species from different orders are represented with different colors: Lepidoptera (bean green), Hemipteran (purple), Diptera (pink), Hymenoptera (orange), Coleoptera (light green), and Neuroptera (yellow). The species abbreviations are provided to identify the respective taxa. Cruf: *C. ruficrus*, Sinv: *Solenopsis invicta*, Amel: *Apis mellifera*, Mmed: *M. mediator*, Acon: *A. confusum*, Mpul: *M. pulchricornis*, Ssp: *S.* sp., Ccun: *Chouioia cunea*, Cmed: *Cnaphalocrocis medinalis*, Dpun: *Dendrolimus punctatus*, Eobl: *Ectropis obliqua*, Cpun: *Conogethes punctiferalis*, Sinf: *Sesamia inferens*, Adis: *Athetis dissimilis*, Bmor: *Bombyx mori*, Dhou: *Dendrolimus houi*, Nlug: *Nilaparvata lugens*, Acer: *Apis cerana cerana*, Tden: *Trichogramma dendrolimi*, Psol: *Phenacoccus solenopsis*, Sfur: *Sogatella furcifera*, Btab: *Bemisia tabaci*, Aluc: *Apolygus lucorum*, Cpal: *Chrysopa pallens*, Tcas: *Tribolium castaneum*, Achi: *Anoplophora chinensis*, Agla: *Anoplophora glabripennis*, Bmin: *Bactrocera minax*, Lsat: *Liriomyza sativae*, Agos: *Aphis gossypii*, Tbra: *Triatoma brasiliensis*, Dant: *Delia antiqua*, Bodo: *Bradysia odoriphaga*, and Azan: *Agrilus zanthoxylumi*.

**Figure 3 insects-14-00920-f003:**
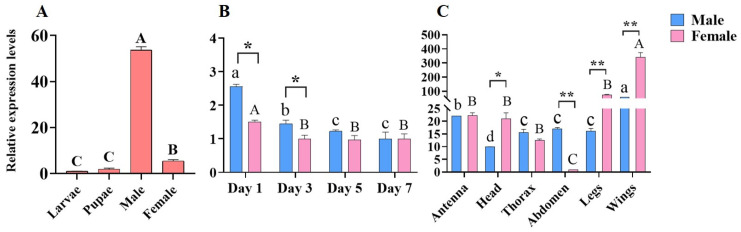
The relative expression levels (mean ± SE) of CrufCSP1 in different developmental stages (**A**), females and males (**B**), and various tissues of *C. ruficrus* (**C**). Different uppercase and lowercase letters indicate significant differences in expression levels between different developmental stages and tissues at *p* < 0.05 according to Tukey’s multiple comparison test. * and ** indicates a significant difference in expression levels between female and male in the same developmental stages and tissues at *p* < 0.05 or *p* < 0.01, respectively.

**Figure 4 insects-14-00920-f004:**
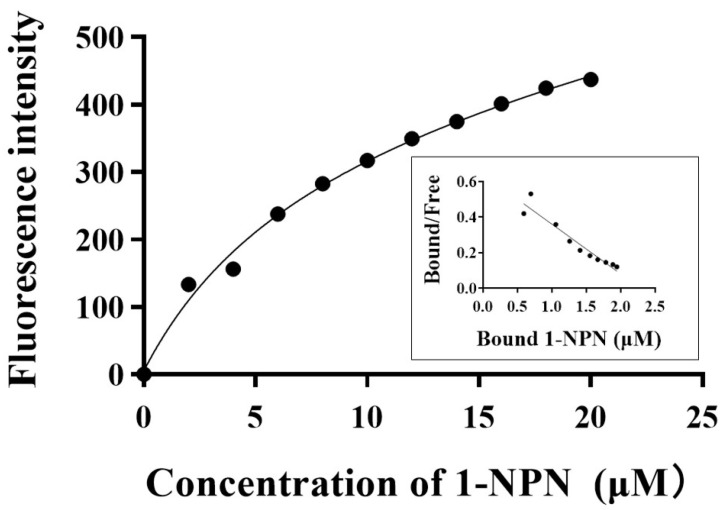
Binding curve for 1-NPN and Scatchard plot of recombinant CrufCSP1.

**Figure 5 insects-14-00920-f005:**
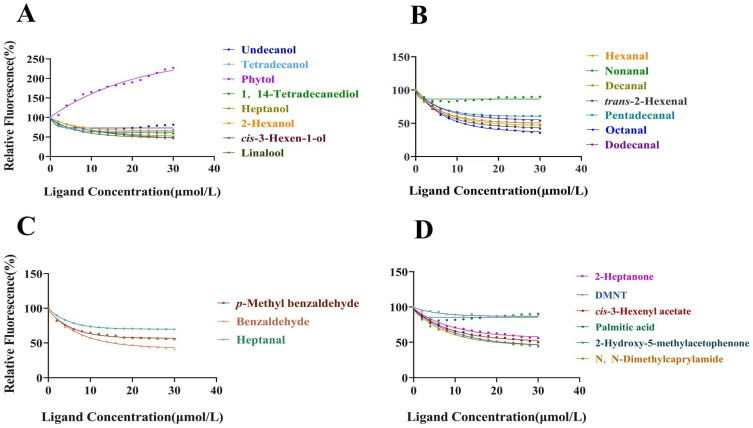
Binding curves of selected ligands to CrufCSP1. (**A**) Alcohol volatiles. (**B**) Aldehyde volatiles. (**C**) Heptanal, benzaldehyde, and *p*-methyl benzaldehyde. (**D**) Other volatiles.

**Figure 6 insects-14-00920-f006:**
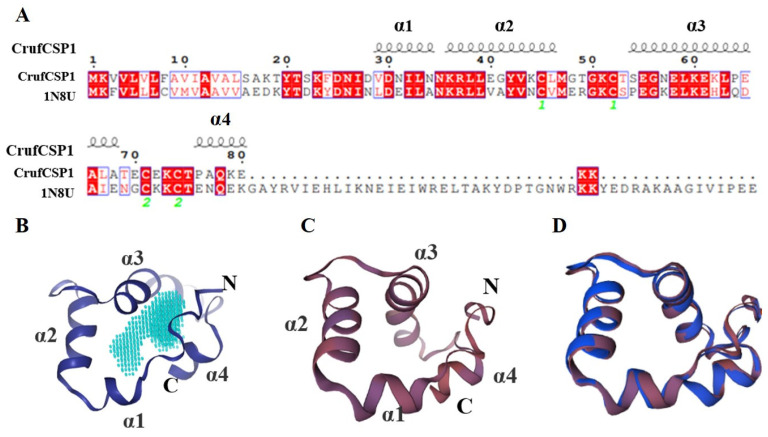
Structural modeling of CrufCSP1. (**A**) Alignment of the sequences of CrufCSP1 and CSPMbraA6 (1N8U), highlighting any similarities or differences between them. (**B**) Three-dimensional structure of CrufCSP1, with the predicted binding pocket highlighted in light blue. The C and N labels indicate the C-terminal and N-terminal regions, respectively. (**C**) Three-dimensional structure of the template protein, 1N8U. (**D**) Superimposed structures of CrufCSP1 (blue color) and CSPMbraA6 (brown color), providing a visual representation of their structural similarities and differences.

**Figure 7 insects-14-00920-f007:**
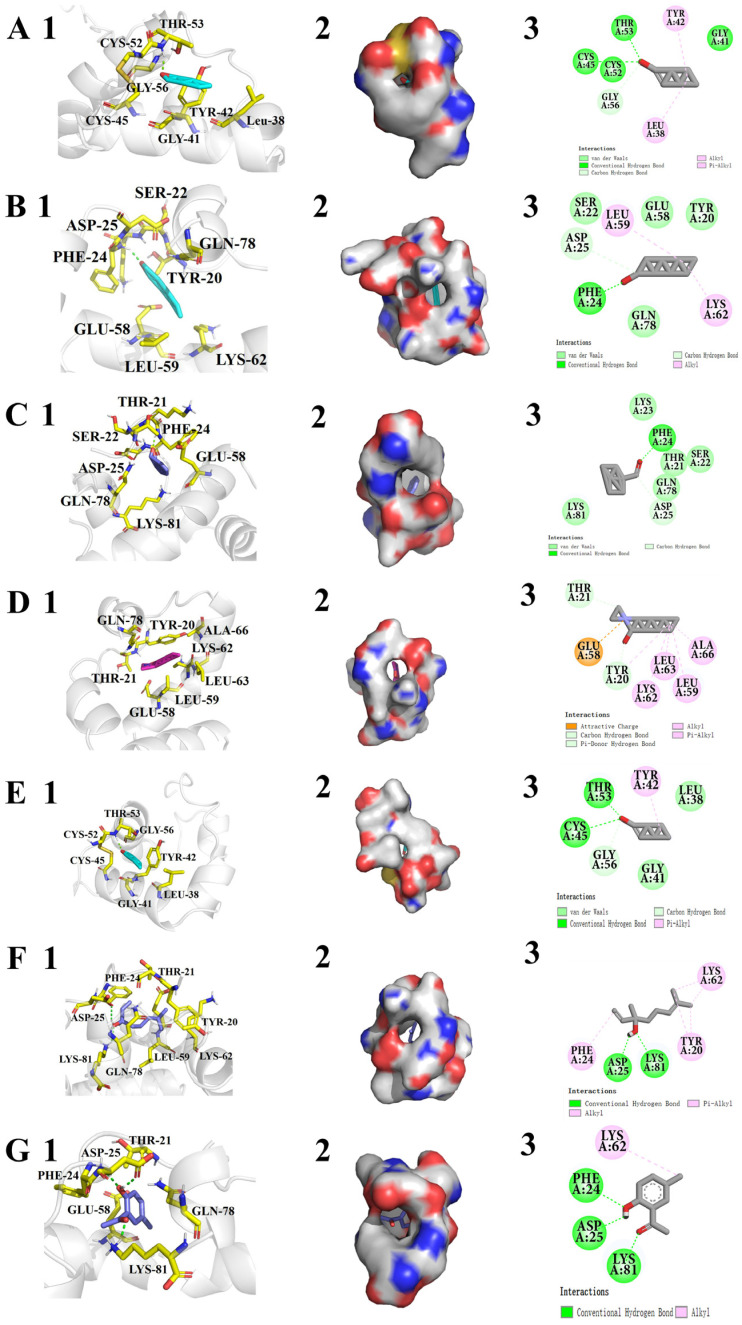
Molecular docking of CrufCSP1 with various ligands. (**A**) *trans*-2-Hexenal, (**B**) Octanal, (**C**) Benzaldehyde, (**D**) *N*,*N*-Dimethylcaprylamide, (**E**) Hexanal, (**F**) Linalool, and (**G**) 2-Hydroxy-5-methylacetophenone. In each panel, we provide different key pieces of information: (1) highlights the key residues of CrufCSP1 involved in the main interactions with the ligand in a 3D representation, and any hydrogen bonds formed are marked with green dot lines; (2) showcases the optimal orientation and conformation of each ligand within the hydrophobic cavity formed by the hydrophobic residues of CrufCSP1; and (3) displays the key residues of CrufCSP1 involved in the main interactions with the ligand in a 2D representation.

**Table 1 insects-14-00920-t001:** The primer sequences of CrufCSP1 for cloning, RT-qPCR, and expression.

Primer Name	Primer Sequence (5′-3′)
For cloning CSP1 open reading frames	
CrufCSP1-Sense	ATGAAGGTCGTCCTAGTTTTG
CrufCSP1-Anti-sense	TTAAACATCAATTCCTTTTGC
For RT-qPCR of CSP1	
CrufCSP1-Sense	TTTTGTTTGCGGTCATAGC
CrufCSP1-Anti-sense	ATTTTCCCGTGCCCATT
β-actin-Sense	GCTCCATCAACCATGAAG
β-actin-Anti-sense	TGGAAGGTGGACAGAGAAG
Heterologous expression of CSP1	
CrufCSP1-Sense	CGCGGATCCATGAAGGTCGTCCTAGTT
CrufCSP1-Anti-sense	CCTCGAGGGCTTTTTTTCTTTCTGTGC

Underlining indicates the sequence of restriction sites.

**Table 2 insects-14-00920-t002:** Chemical compounds in the binding assays of recombinant protein CrufCSP1 of *C. ruficrus*.

Ligand	Formula	CAS	Purity	Source	IC_50_ (μM)	K_i_ (μM)
Body volatiles						
Undecanol	C_11_H_24_O	112-42-5	98%	Macklin	>50	>50
Tetradecanol	C_14_H_30_O	112-72-1	99.5%	Macklin	>50	>50
1,14-Tetradecanediol	C_14_H_30_O_2_	19812-64-7	98%	Macklin	>50	>50
2-Hexanol	C_6_H_14_O	626-93-7	98%	Energy Chemical	>50	>50
Phytol	C_20_H_40_O	150-86-7	90%	Macklin	>50	>50
Dodecanal	C_12_H_24_O	112-54-9	95%	Macklin	41.06	35.96
Pentadecanal	C_15_H_30_O	629-62-9	98%	Aladdin	44.48	38.96
*p*-Methyl benzaldehyde	C_8_H_8_O	104-87-0	97%	Macklin	45.96	40.26
Palmitic acid	C_16_H_32_O_2_	57-10-3	97%	TCL	>50	>50
2-Hydroxy-5-methylacetophenone	C_9_H_10_O_2_	1450-72-2	98%	Macklin	22.79	19.97
*N*,*N*-Dimethylcaprylamide	C_10_H_21_NO	1118-92-9	95%	Macklin	22.02	19.29
Corn volatiles						
Hexanal	C_6_H_12_O	66-25-1	99%	Sigma	22.10	19.36
Heptanal	C_7_H_14_O	111-71-7	99%	Macklin	>50	>50
Octanal	C_8_H_16_O	124-13-0	99.5%	Macklin	17.17	15.04
Nonanal	C_9_H_18_O	124-19-6	99%	Macklin	>50	>50
Decanal	C_10_H_20_O	112-31-2	99%	Aladdin	24.90	21.81
Benzaldehyde	C_7_H_6_O	100-52-7	98%	Sigma	17.39	15.24
*trans*-2-Hexenal	C_6_H_10_O	6728-26-3	90%	TCL	16.54	14.49
Heptanol	C_7_H_16_O	111-70-6	98%	Sigma	31.56	27.65
*cis*-3-Hexen-1-ol	C_6_H_12_O	928-96-1	97%	Sigma	28.23	24.73
Linalool	C_10_H_18_O	78-70-6	98%	Sigma	22.60	19.80
DMNT	C_11_H_18_	19945-61-0	97%	Yuanye bio-technology	>50	>50
*cis*-3-Hexenyl acetate	C_8_H_14_O_2_	3681-71-8	98%	Sigma	33.36	29.22
2-Heptanone	C_7_H_14_O	110-43-0	95%	Sigma	>50	>50

**Table 3 insects-14-00920-t003:** Docking results for CrufCSP1 with seven ligands.

Ligand	Binding Energy (Kcal/mol)	Residues Involved in Hydrogen Bonds	Close-Contact Interacting Residues
*trans*-2-Hexenal	−3.68	Thr53 (1.9 Å)	Leu38, Gly41, Tyr42, Cys45, Cys52, and Gly56
Octanal	−3.77	Phe24 (1.9 Å)	Tyr20, Ser22, Asp25, Glu58, Leu59, Lys62, and Gln78
Benzaldehyde	−4.08	Phe24 (1.9 Å)	Thr21, Ser22, Lys23, Asp25, Glu58, Gln78, and Lys81
*N*,*N*-Dimethylcaprylamide	−4.65	-	Tyr20, Thr21, Glu58, Leu59, Lys62, Leu63, Ala66, and Gln78
Hexanal	−3.52	Thr53 (2.0 Å)	Leu38, Gly41, Tyr42, Cys45, Cys52, and Gly56
Linalool	−5.01	ASP25 (2.2 Å) LYS81 (2.4 Å)	Tyr20, Thr21, Phe24, Glu58, Leu59, Lys62, and Gln78
2-Hydroxy-5-methylacetophenone	−5.24	Phe 24 (2.5 Å) ASP25 (2.0 Å) LYS81 (1.9 Å)	Thr21, Glu58, and Gln78

## Data Availability

Data are contained within the article or Appendix A.

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
