# Peer review of "Characterization of CrufCSP1 and Its Potential Involvement in Host Location by Cotesia ruficrus (Hymenoptera: Braconidae), an Indigenous Parasitoid of Spodoptera frugiperda (Lepidoptera: Noctuidae) in China"

_insects, 2023, doi:10.3390/insects14120920_

Round 1

Reviewer 1 Report

Comments and Suggestions for Authors

See attach

Comments on the Quality of English Language

Minor editing of English language is required

Author Response

Dear Reviewer,

I would like to take this opportunity to express my sincere gratitude for your time and efforts in reviewing my article. Your positive feedback and encouragement have been instrumental in enhancing the quality of the final manuscript, and I am truly honored to have had the benefit of your expertise and insights.

The point to point responds to your comments were finished. We greatly appreciate your warm work earnestly, and hope the correction will meet with approval.

Best regards,

Zhang S.Z.

Reviewer 2 Report

Comments and Suggestions for Authors

The manuscript titled "Characterization of CrufCSP1 and its Potential Involvement in Host Location in Cotesia ruficrus (Hymenoptera: Braconidae) and Indigenous Parasitoids of Spodoptera frugiperda (Lepidoptera: Noctuidae) in China" is well-structured, and the methodology employed for characterization and affinity studies is adequate. However, in order to enhance the clarity and completeness of this significant study prior to publication, certain key points need clarification.

It is necessary to provide a clear rationale for selecting CrufCSP1 for this study. The manuscript cite a study (reference 30, (Wang et al, 2023) that was not published yet, therefore we cannot have access about how was identified this CrufCSP1 and why it was chosen for investigation.. Was this the only one CSP identified in Cotesia ruficrus (Hymenoptera: Braconidae)? This could be clarified in the introduction. Furthermore,  it is noteworthy that CrufCSP1,in contratst to other CSPs, possesses only four cysteine residues. Typically, CSPs exhibit six cysteines, and there is report of five cysteine, forming five to seven alpha-helices in odorant binding proteins. This intriguing distinction warrants discussion within the manuscripts introduction and discussion sections. The authors could explore into the significance of this unique characteristic and elaborate on how it might influence the functional properties of CrufCSP1. Additionally, providing more information about other CSPs identified in C ruficrus would contribute to a more comprehensive understanding of the context and significance of CrufCSP1 in the species.  

Material and Methods

Line 107. Change breeding for  rearing

Line 113 Delete “any observed”.

The authors should include details regarding the specific larval instar used, the age of the pupae, and whether an equal number of male and female pupae were employed in the study. This information is particularly crucial for the pupal stage, given that males tend to produce a greater quantity of CSP1 compared to females. Using an uneven ratio of male to female pupae could potentially obscure the results, making it essential to clarify the distribution of pupal genders in the experimental setup.

Please add the age of the insects that was used to remove the tissues. Was the same age of the extracts of the total body?

Discussion

Line 436-437 The authors propose that the elevated levels of CSP1 in males may be attributed to their need to locate females for mating, I would like to suggest to authors add some information about if  females also require effective means to find males for mating and identify suitable hosts for egg laying. A more in-depth exploration of these aspects in the discussion would enrich the manuscript.

Furthermore, it would be beneficial for the authors to support their argument by referencing studies that delve into the mating behaviour of C. ruficrus. Specifically, incorporating a reference that elucidates how males follow semiochemicals to locate females for mating would strengthen the connection between the observed CSP1 levels and the species' reproductive strategies. This additional information would contribute to a more comprehensive understanding of the intricate ecological dynamics involved in the behaviour of C. ruficrus.

Lines 457-458 the authors describe that CrufCSP1 demonstrated better binding capabilities with most corn volatiles than with body volatiles. Interesting, all volatile organic compounds (VOCs) demonstrating affinity with CSP1 were characterized by their polarity and a carbon  structure of less than 10. Some important compounds for natural enemies such as DMNT and TMTT in their foraging behaviours for hosts and prey were not bound by CrufCSP1 (Magalhães et al., 2016; Gurr et al., 2023). This observation suggests the likelihood of other CSPs playing a role in the perception of crucial plant volatiles and apolar compounds.

Gurr, FM., Liu J., Pickett J.A & Stevenson P.C. (2023). Review of the Chemical ecology of homoterpenes in arthropod plant interactions. Austral Entomology,  61 (1):3-14.

Magalhães, D.M., Borges, M., Laumann, R.A., Woodcock, C.M., Pickett, J.A., Birkett, M.A. & Blassioli-Moraes, M.C. (2016). Influence of two acyclic homoterpenes (Tetranorterpenes) on the foraging behavior of Anthonomus grandis Boh. Journal of Chemical Ecology, 42:305-313.

Author Response

(The authors gave the same response as above.)

Reviewer 3 Report

Comments and Suggestions for Authors

This is a reverse chemical ecology study, oriented to elucidate the importance of CrufCSP1 protein for the location of Spodoptera frugiperda by the parasitoid Cotesia ruficrus. Although the results of this research do not allow us to conclude that CrufC SP1 plays an important role in the direct recognition of the host, it was determined that this protein might be relevant in the detection of compounds associated with corn (an important resource for the host) by the parasitoid. The information generated contributes to understanding the various roles that OBPs could play in insects, particularly in C. ruficrus. This information could eventually also be relevant for the design of management strategies for S. frugiperda.

This work can be published after making some minor adjustments. There are some aspects that authors are recommended to review, listed below:

Line 71: Explain abbreviations the first time they are used: Fall Armyworm (FAW)

Line 178: Tests of normality and homogeneity of variances are important for validating ANOVA assumptions. If the authors made them, they should mention them here.

Line 283 and 323: Scientific names must be in italics. 

Figure 3: Error bars are standard error of mean or standard deviation?

Figure 3b: The letters suggest that days within the same sex were compared and there was no comparison between the two sexes. Clarify this point.

Figure 3c: The letters suggest that tissues within the same sex were compared and there was no comparison between the two sexes. Clarify this point.

Figure 3 legend: ....stages and tissues at P < 0.05 add "according to Tukey's multiple comparison test. "

The comments are also in the attached file

Comments on the Quality of English Language

Minor editing of the English language required

Author Response

(The authors gave the same response as above.)
